# Fractionated Stereotactic Intensity-Modulated Radiotherapy for Large Brain Metastases: Comprehensive Analyses of Dose–Volume Predictors of Radiation-Induced Brain Necrosis

**DOI:** 10.3390/cancers16193327

**Published:** 2024-09-28

**Authors:** Taro Murai, Yuki Kasai, Yuta Eguchi, Seiya Takano, Nozomi Kita, Akira Torii, Taiki Takaoka, Natsuo Tomita, Yuta Shibamoto, Akio Hiwatashi

**Affiliations:** 1Department of Radiation Oncology, Shonan Kamakura General Hospital, 1370-1 Okamoto, Kamakura 247-8533, Kanagawa, Japan; 2Department of Radiology, Nagoya City University Graduate School of Medical Sciences, 1 Kawasumi, Mizuho-ku, Nagoya 467-8601, Aichi, Japan; stakano@med.nagoya-cu.ac.jp (S.T.); a68553@med.nagoya-cu.ac.jp (N.K.); atorii@med.nagoya-cu.ac.jp (A.T.); xcjh3432@med.nagoya-cu.ac.jp (T.T.); ntomita@med.nagoya-cu.ac.jp (N.T.); ahiwata@med.nagoya-cu.ac.jp (A.H.); 3Department of Radiology, Nagoya City University Hospital, 1 Kawasumi, Mizuho-cho, Mizuho-ku, Nagoya 467-8602, Aichi, Japan; raykasai@med.nagoya-cu.ac.jp (Y.K.); ra2580@med.nagoya-cu.ac.jp (Y.E.); 4Narita Memorial Proton Center, 78 Shirakawa-cho, Toyohashi 441-8021, Aichi, Japan; yshiba@meiyokai.or.jp

**Keywords:** brain necrosis predictor, fractionated stereotactic radiotherapy, large brain metastasis, comprehensive analysis, multicollinearity

## Abstract

**Simple Summary:**

To examine predictors of brain necrosis (BN) in fractionated SRT for BMs, comprehensive analyses of dosimetric parameters were conducted. The normal brain volume receiving an xx Gy biological dose in 2 Gy fractions (VxxEQD2) was calculated, and collinearities were carefully excluded. Three-fraction SRT was delivered to 34 patients with 75 BMs, five-fraction SRT to 58 patients with 104 BMs, and ten-fraction SRT to 20 patients with 37 BMs depending on the target volume. The incidence of BN was also significantly higher in cases with V55EQD2 > 30 cc or V60EQD2 > 20 cc. These doses correspond to 28 or 30 Gy/5 fr and 37 or 40 Gy/10 fr, respectively. These biologically high doses may affect BN incidence. Thus, the planning target volume margin should be minimized as much as possible.

**Abstract:**

Background: The objective was to explore dosimetric predictors of brain necrosis (BN) in fractionated stereotactic radiotherapy (SRT). Methods: After excluding collinearities carefully, multivariate logistic models were developed for comprehensive analyses of dosimetric predictors in patients who received first-line fractionated SRT for brain metastases (BMs). The normal brain volume receiving an xx Gy biological dose in 2 Gy fractions (VxxEQD2) was calculated from the retrieved dose–volume parameters. Results: Thirty Gy/3 fractions (fr) SRT was delivered to 34 patients with 75 BMs (median target volume, 3.2 cc), 35 Gy/5 fr to 30 patients with 57 BMs (6.4 cc), 37.5 Gy/5 fr to 28 patients with 47 BMs (20.2 cc), and 40 Gy/10 fr to 20 patients with 37 BMs (24.3 cc), according to protocols, depending on the total target volume (*p* < 0.001). After excluding the three-fraction groups, the incidence of symptomatic BN was significantly higher in patients with a larger V50EQD2 (adjusted odds ratio: 1.07, *p* < 0.02), V55EQD2 (1.08, *p* < 0.01), or V60EQD2 (1.09, *p* < 0.01) in the remaining five- and ten-fraction groups. The incidence of BN was also significantly higher in cases with V55EQD2 > 30 cc or V60EQD2 > 20 cc (*p* < 0.05). These doses correspond to 28 or 30 Gy/5 fr and 37 or 40 Gy/10 fr, respectively. Conclusions: In five- or ten-fraction SRT, larger V55EQD2 or V60EQD2 are BN risk predictors. These biologically high doses may affect BN incidence. Thus, the planning target volume margin should be minimized as much as possible.

## 1. Introduction

The management of newly diagnosed brain metastases (BMs) encompasses a spectrum of therapeutic modalities, including surgical interventions, radiotherapy, and other adjunctive treatments [1,2,3]. Among radiotherapy techniques, stereotactic radiotherapy (SRT) is commonly delivered in a single fraction (radiosurgery), which achieves excellent local control (LC) with low toxicity [1,2,3]. Brain necrosis (BN) is a well-characterized adverse effect of stereotactic radiotherapy (SRT) and is occasionally associated with serious neurological sequelae. Brain necrosis encompasses a broad clinical spectrum; patients may present with an incidental imaging finding in the absence of symptoms or with symptoms including neurological deficits, headaches, and seizures [4]. Many risk factors have been implicated in the development of BN, including the prescribed radiation dose, reirradiation, treated volume, histology, and the use of concurrent systemic therapies [4], including immunotherapy or targeted therapy. These have almost exclusively been validated in patients treated with radiosurgery for small lesions [5].

The target volume is historically considered a predictor for BN in radiosurgery, and this approach is not regarded as suitable for patients with tumors larger than 2–3 cm at the maximum diameter. A higher incidence of radiation-induced brain necrosis (BN) after radiosurgery was reported than fractionated SRT [5,6,7]. On the other hand, fractionated SRT for large BMs aims to deliver highly conformal treatments while improving LC and decreasing neurotoxicity through potential dose escalations and the inter-fraction repair of normal tissues [8,9]. Minniti et al. [5] found that fractionated SRT (27 Gy/3 fx) might reduce BN incidence compared to SRS (8% vs. 20%, *p* = 0.004). Few studies have investigated the effects of the fractionated SRT dose, fractionation, and target volume on the risk of BN [9,10,11,12,13,14]. Dose–volume predictors in fractionated SRT have been reported by fewer studies compared with radiosurgery [7,12,13]. The current HyTEC report recommends that the dose constraints are V20Gy < 20 cc in three-fraction SRT and V24Gy < 20 cc in five-fraction SRT [12,13]. These reports were based on multivariate analyses of retrospective observation studies of small lesions. Recent statistical advances recommend that exploring dosimetric predictors should take multicollinearity into account [15]. Multicollinearity occurs when two or more variables strongly correlate with each other, making it difficult to develop an accurate model. Therefore, dosimetric parameters in fractionated SRT for large BMs should be evaluated by considering these statical factors.

Immunotherapy or targeted therapy improves overall survival significantly [16,17,18]. Molecular agents targeting these mutations improve OS significantly in advanced-stage patients with lung cancers, liver cancers, gastric cancer, and breast cancers [19]. The randomized trial, DESTINY-Breast04, revealed that trastuzumab–deruxtecan, an antibody–drug conjugate consisting of an anti-HER2 monoclonal antibody linked to a topoisomerase I inhibitor, improves PFS and OS for patients with HER2-low metastatic breast cancer [20]. In addition, immune checkpoint inhibitors prolong the prognosis of patients with melanoma, colorectal cancer, cervical cancer, or lung cancer. The CheckMate 067 trial showed the benefit of dual checkpoint inhibition over monotherapy with a CTLA-4 inhibitor. With a minimum follow-up of 10 years, median OS was 71.9 months with nivolumab plus ipilimumab, 36.9 months with nivolumab, and 19.9 months with ipilimumab [21]. BMs are increasingly managed with a combination of SRT and these systemic therapies for extracranial disease control. This shift in today’s practice necessitates a careful SRT delivery to avoid BN. The purpose of the present study was to explore potentially modifiable dose–volume metrics that predict BN after fractionated SRT.

## 2. Materials and Methods

### 2.1. Study Design and Patients

Patients treated with SRT for BMs at a single institution between 2012 and 2021 were reviewed. SRT for BMs was performed according to prospective protocols [22]. Patients who fulfilled the following inclusion criteria were treated with SRT: (1) World Health Organization performance status of 0–2, (2) patient conditions allowing the same body position in an immobilizing device for more than 20 min, and (3) BM number ≤ 10. Exclusion criteria were as follows: (1) previous surgery or a history of radiotherapy for BM, (2) meningitis carcinomatosa, (3) pregnancy or potential pregnancy, (4) psychiatric disorders, and (5) contraindication to iodine or gadolinium contrast. Informed consent for the SRT was obtained from all patients or their guardians. This retrospective study was approved by the Institutional Review Board of Nagoya City University (No. 60-19-0207).

### 2.2. Radiotherapy Details

Patients were placed in a supine position and a thermoplastic mask was molded to the head and attached to the head support. The planning target volume (PTV) denoted a visible tumor on computed tomography (CT) and contrast-enhanced magnetic resonance imaging (MRI) plus 0–2 mm depending on nearby critical organs. Computed tomography (CT) was acquired and fused with contrast-enhanced magnetic resonance imaging (MRI). The sequence was axial T1-weighted imaging with 1 mm slice-thickness mainly acquired with 1.5-T machines. The clinical target volume was defined as an abnormal contrast-enhanced lesion on the MRI. The PTV was the target plus 0–2 mm margin depending on the nearby critical structures, such as the optic nerve, brain stem, and so on. These margins were reduced in cases where critical organs were nearby targets.

In the first protocol employed between 2012 and 2016, the basic prescribed dose was 35 Gy/5 fractions (fr). A dose of 30 Gy/3 fr was permitted for small lesions (maximum diameter <1.5 cm), while 37.5 Gy/5 fr was used for large lesions (≥3 cm). Since SRT for large total PTV (≥15 cc) was associated with a risk of neurotoxicity, we employed different fractionation protocols for these tumors [22]. Between 2017 and 2021, the protocol for large total PTV was revised to reduce the incidence of BN. Treatment for large total PTV(cc) (≥15 cc) with 35 Gy or 37.5 Gy/5 fr was superseded by 40 Gy/10 fr. In the new protocol, 30 Gy/3 fr for small lesions and 35 Gy/5 fr for other lesions remained. The prescribed dose covered at least 50% of PTV. At least 90% of the dose covered 95% of the PTV.

Dose constraints were applied to adjacent structures; the maximum doses to the brain stem, optic nerve, and optic chiasm were limited to <36 Gy/10 fr, 25 Gy/5 fr, or 18 Gy/3 fr [22]. To satisfy these limitations, the simultaneous integrated boost technique is used to reduce critical organ doses. The SRT was delivered with the helical mode in tomotherapy Hi-ART system^®^ or Radixact^®^. Treatment was performed three times a week to efficiently utilize reoxygenation phenomena [23]. Any systematic agent, including immunotherapy and tyrosine kinase inhibitors (TKI), was not allowed 1 week before and after these treatments to avoid increased toxicity [1,4].

### 2.3. Patient Follow-Ups, Endpoints, and Dose–Volume Parameters

Patients were followed every 2 to 3 months after SRT with contrast-enhanced MRI and physical examinations. Local recurrence was defined as a ≥20% increase in the sum of diameters of target lesions of the contrast-enhanced tumor on MRI or CT, taking as reference the smallest sum in the study. These evaluations were based on RECIST v1.1 [24].

BN was pathologically verified or diagnosed by perfusion and magnetic resonance spectroscopy (fMRI) or C-11 methionine positron emission tomography (MetPET) [25,26,27]. Recurrence in MetPET was diagnosed based on the mean tumor/cortex ratio values (>2.0). Lesions with N-acetyl aspartate/choline ratio <1 or lipid/choline ratio <3 in the fMRI or relative cerebral blood volume of >2.1 were considered as recurrence. These were diagnosed by diagnostic radiologists with >5 years of neuroradiology experience. If no radiological or pathological evidence of recurrence is observed, the lesion can be classified as BN.

Toxicities were recorded according to the Common Terminology Criteria for Adverse Events (CTCAE) v.4.0, the Japanese Clinical Oncology Group version. 

To examine dose–volume predictors of BN, the physical dose to the targets and the normal brain (VxGy) were extracted for every 1 Gy. The conformity index (CI) and uniformity index (UI) were calculated according to the following formulae [28].
Uniformity index (UI) = D5%/D95%
Conformity index (CI) = (V_PTV_/TV_PV_)/(TV_PV_/V_TV_)

Abbreviations in these formulae are as follows: V_PTV_ = PTV (cc), TV_PV_ = lesion volume (cc) covered by the prescribed isodose, V_TV_ = prescribed isodose volume (cc), and D5% = minimum dose delivered to 5% of PTV. Lower CI indicates higher conformity, while lower UI indicates better homogeneity. Ideal CI and UI are both 1. 

Biological equivalent doses (BED) to the brain with an α/β ratio of 2 Gy were calculated using the linear-quadratic model to assess different fractionation schedules [23]. A conversion table after rounding figures in each schedule is shown in Table 1. In this analysis, the normal brain (the brain minus visible tumors) volume receiving xx Gy BED in 2 Gy fractions was described as VxxEQD2. 

### 2.4. Statistical Analysis

Fisher’s exact test or a one-way analysis of variance was applied to compare categorical or continuous variables. Overall survival (OS), LC, and symptomatic BN (grade 2) probabilities were evaluated using the Kaplan–Meier method from the start of SRT. The cumulative incidence of local recurrence was calculated, accounting for death as a competing risk. Death and local recurrence were assumed as risks of BN. 

A logistic regression analysis (LRA) was conducted to identify BN dosimetric predictors. Multicollinearity in dose–volume parameters introduces errors in a multivariate regression analysis. Collinearity was tested using the coefficient of determination (R^2^) between each dose–volume parameter, assuming collinearity as >0.4 R^2^ in this study [15,29]. The sample size was calculated with Smeden’s formula [30,31]. Given 2–4 candidate predictors and an outcome proportion of 0.08, a sample size of at least 56–105 participants was required to target a mean absolute error of 0.05 between observed and true outcome probabilities. 

With deviations from the linear-quadratic model in a high dose per fraction schedule [23], VxxEQD2 in the 3-fraction group may not correspond biologically to that in the 5- or 10-fraction group. In other words, the EQD2 may be underestimated compared to the actual biological dose. Therefore, the final LRA was also conducted after excluding the 3-fraction group. 

All analyses were performed in EZR v1.41, which is a graphical user interface for R v3.6.1 (The R Foundation for Statistical Computing, Vienna, Austria) [32].

## 3. Results

### 3.1. Patient Characteristics, Treatment Details, and Outcomes

Patient and treatment details are summarized in Table 2. In total, 112 patients with 215 BMs were treated with these protocols. Among them, 76 patients (68%) had lung cancers and 106 had extracranial lesions. Thirty Gy/3 fractions (fr) SRT was delivered to 34 patients with 75 BMs (median target volume, 3.2 cc), 35 Gy/5 fr to 30 patients with 57 BMs (6.4 cc), 37.5 Gy/5 fr to 28 patients with 47 BMs (20.2 cc), and 40 Gy/10 fr to 20 patients with 37 BMs (24.3 cc), respectively (*p* < 0.00001). Of 216, 32 individual BM volumes were ≥15 cc. Five BMs were located in the brainstem, four in the thalamus, and two in the basal ganglia. Seventy BMs (33%) were in the frontal cortex or subcortex, thirty-nine (18%) in the parietal lobe, thirty-six (17%) in the occipital lobe, thirty-five (16%) in the temporal lobe, and twenty-five (12%) in the cerebellum. Age, sex, extracranial disease, primary cancer, CI, and UI did not significantly differ between these four groups (*p* > 0.18) (Table 2). As the three-fraction SRT was used for smaller lesions, the CI in the three-fraction SRT’s variance was wide but not significant (*p* > 0.36).

OS and LC curves are shown in Figure 1a,b. The 1-year OS rate was 54% (median, 13 months). During the follow-up period, new BMs developed in 36 patients. One-year LC rates were 92% in the three-fraction group, 94% in the five-fraction group, and 82% in the ten-fraction group (Gray’s test, *p* = 0.11). 

### 3.2. Toxicities

Grade 2 seizures were observed in eight patients and a grade 2 headache was observed in one patient. Grade 5 and 1 intratumor bleeding occurred in one patient each. The brain CT of the grade 5 patient showed intratumor bleeding with brain herniation irradiated with SRT. Thus, the bleeding was evaluated as caused by SRT. BN (≥grade 2) developed in four patients in the three-fraction group, seven patients in the five-fraction group, and zero patients in the ten-fraction group. Among them, five cases were pathologically confirmed following craniotomy. Two cases were diagnosed using MetPET, and the other four by fMRI and were verified after the follow-up. Grade 1 BN was observed in three patients in the three-fraction group, two patients in the five-fraction group, and one patient in the ten-fraction group. Grade 1 BN was asymptomatic and diagnosed with fMRI and local recurrence was not suspected in any case after careful follow-ups. The incidence of BN (≥grade 2) at 1 year in the Kaplan–Meier method was 9% in the three-fraction group, 8% in the five-fraction group, and 0% in the ten-fraction group (*p* = 0.29) (Figure 1c).

### 3.3. Multivariate Analyses of BN

Figure 2 shows how each dosimetric variable is correlated to one another. Strong multicollinearity (R^2^ ≥ 0.42) appeared between each dose. Therefore, PTV (cc), CI, UI, and each VxxEQD2 (cc) were included as continuous variables in LRA to examine the models [33,34]. In analyses of 112 patients, no significant parameter was identified (Figure 3a,b).

The three-fraction group was excluded from these LRAs due to the biological imprecision of EQD2 in the fractionation. In addition, the sample size of the group was not sufficient for independent analyses. Therefore, the LRA was conducted after excluding the three-fraction group. In the analyses, the incidence of BN was higher in patients with larger V50EQD2, V55EQD2, and V60EQD2 (Figure 3c,d). Adjusted odds ratios for ≥grade 1 and ≥grade 2 BN were 1.06 (95% confidence interval: 1.01–1.12, *p* < 0.03) and 1.06 (1.01–1.12, *p* < 0.02) for V55EQD2 (cc), respectively (Appendix A). Similar results were observed in cases with larger V50EQD2 and V60EQD2. In consideration of previously reported sample sizes and risks [4], PTV (cc) and each VxxEQD2 (cc) were included in the final LRAs (Table 3). V50-60EQD2 appeared to be a significant predictor of grades 1 and 2 BN.

The actual incidences of BN are depicted in Figure 4 and Appendix A to explore the cutoff value in each VxxEQD2. These PTV(cc) and VxxEQD2(cc) were categorized into three groups on an ad hoc basis. The adjusted odds ratios of BN (≥grade 2) were significantly higher with V60EQD2 >20 cc (*p* = 0.04) or V55EQD2 >30 cc (*p* = 0.04) (Table 4 and Appendix A).

## 4. Discussion

The incidence of BN is recognized as the dose-limiting toxicity of SRT for BMs. The major backgrounds are emerging systematic agents and improvements in the prognosis of patients with advanced-stage cancer [35]. For example, the systemic treatment of advanced non-small cell lung cancer remained conventional chemotherapy for several decades until the emergence of TKI therapy following the discovery of several driver mutations including EGFR, ALK, ROS1, MET, RET, BRAF, and ERBB2/HER2 [1,17,18]. Molecular agents targeting these mutations improve OS significantly in advanced-stage patients with lung cancers, liver cancers, gastric cancer, and breast cancers [19]. In addition, immune checkpoint inhibitors prolong the prognosis of patients with melanoma, colorectal cancer, cervical cancer, or lung cancer. In the landmark trial, KEYNOTE-407, 559 patients with advanced squamous cell lung cancer were randomized to chemotherapy with and without pembrolizumab. Outcomes were improved with pembrolizumab plus chemotherapy; the 5-year OS rates were 18% in the pembrlizumab arm and 9.7% in the control arm, respectively [36]. An interaction between these new agents and BN has not been established, while some reports suggest that immune-checkpoint inhibitors increase BN incidence [3]. Therefore, dose constraints in SRT should be more prudent and conservative in this immunotherapy era. 

The second reason why BN is a major topic is the extension of SRT adaptation. To preserve neurocognitive function and reduce the treatment period, SRT was delivered to multiple or larger BMs [2,9,10,11,12,37,38]. Even in pre-operative and post-operative settings, SRT is getting to play an indispensable role instead of whole-brain radiotherapy [4,11,39]. In post-operative SRT, the target, resected cavity is generally larger than the pre-operative visible tumor. The target volume is regarded as a risk factor for BN. Thus, the dose-limiting toxicity of treatment is BN, fractionation is used in these settings, and dose constraints in fractionation can be an argumentative issue. 

In single-fraction SRT, V12Gy of the normal brain is canonically regarded as a dose-limiting indicator based on clinical data for small BMs [14]. While large lesions are often treated with fractionated SRT in actual clinical settings [6,9,40,41], dose–volume predictors of the normal brain have not yet been established. Upadhyay et al. [42] reported the largest cohort outcomes for 434 patients with 2518 lesions treated with three-fraction SRT, but these lesions were mainly small (mean volume, 2.6 cc). The present study conducted comprehensive dose–volume analyses of BM patients receiving fractionated SRT for larger tumors. In the five- and ten-fraction groups, normal brain volume receiving high BED was correlated with higher BN incidence. The incidence of BN was less than 8% for V60EQD2 ≤ 20 cc or V55EQD2 ≤ 30 cc. These doses correspond to 28 or 30 Gy/5 fr and 37 or 40 Gy/10 fr, respectively. Therefore, these results suggest that the following dose constraints should be at least maintained: V28Gy < 30 cc or V30 < 20 cc in five-fraction SRT and V37Gy < 30 cc or V40 < 20 cc in ten-fraction SRT. In addition, this result implies that the PTV margin should be minimized as much as possible to reduce high BED volume. 

The latest guidelines [2,5,7,12] suggest provisional dose constraints of the normal brain in fractionated SRT for BMs. In the consensus statement, V25Gy, V28.8Gy, and V30 of the normal brain in five-fraction SRT cannot exceed 16, 7, or 10.5–30 cc, respectively. This recommendation is based on two clinical studies. Inoue et al. [43] examined 85 BMs in 78 patients. There were 16 lesions with V28.8Gy ≥7.0 cc, and two developed extensive brain edema due to BN. None of the patients with V28.8Gy < 7.0 cc developed edema that required surgical intervention. Andreaska et al. [44] conducted a multi-institutional retrospective review of 117 BMs in 83 patients treated with five-fraction SRT. In lesions without prior SRT, V25Gy > 16 cc and V30Gy > 10 cc were associated with a significantly higher risk of BN. Although these findings provide insights into dose–volume predictors, these reports do not mention collinearity between parameters and the reason why these parameters are included. Multicollinearity is a statistical phenomenon characterized by strong correlations or dependencies among predictor variables in a regression model [15]. It occurs when two or more variables strongly correlate with each other, making it difficult for the model to differentiate the individual effects of each variable on the dependent variable. Errors stemming from violations of the multicollinearity assumption are relevant to radiation dose–volume research. Due to strong correlations among variables derived from points along individual organ dose–volume histogram curves, dose–volume analyses are susceptible to multicollinearity errors. The present study analyzed dose–volume parameters comprehensively in consideration of the multicollinearity of each parameter, suggesting that higher BED rather than lower BED affected the incidence of BN in five- or ten-fraction SRT. The result partially supports the guidelines’ recommendations [2,5,7,12]. 

In the context of the pathophysiology of BN, there are two main theories: (i) glial cell damage and (ii) vascular injury [4]. In the first scenario, radiation may also damage glial cells. Radiation-induced cell damage leads to the accumulation of double-strand deoxyribonucleic acid (dsDNA) in the cytosol of tumor, stromal, endothelial, and immune cells, activating the cGAS-STING pathway [35,45]. In this pathway, cGAS, an enzyme that recognizes cytosolic dsDNA, induces the up-regulation of type 1 interferons and dendritic cell activation, ultimately triggering various inflammatory effector responses. A higher radiation dose, to a certain degree, induced the greater accumulation of dsDNA in the cytosol. Therefore, it is reasonable that higher BED has potential as a dose–volume predictor.

In contrast to five- and ten-fraction SRT outcomes, BN predictors were not clarified in three-fraction SRT in the present study. These results are partially explained by linear-quadratic model limitations. The model fits well if a single-fraction dose was less than 2-fold of the organ α/β ratio. With a higher dose per fraction, the quadratic cell-killing component dominates in the model, and the deviation becomes evident [23]. Therefore, VxxEQD2 in the three-fraction group may not correspond biologically to that in the five- or ten-fraction group. In addition, radiation disrupts the blood–brain barrier, resulting in increased capillary leakiness and vascular permeability in the second scenario of the BN pathophysiology [4,45]. Radiation, particularly in large fraction sizes >8 Gy, activates acid sphingomyelinase and induces the up-regulation of ceramide, which causes anarchic vessel sprouting resulting in ischemia and cell death. These pathologies of BN in the three-fraction group may be different from those in the five- and ten-fraction groups.

There are several limitations in this study. One of the most controversial issues is about BN diagnosis. In this study, five of eleven BN cases (≥grade 2) were pathologically confirmed. On the other hand, two were diagnosed using MetPET, while the other four cases with BN (≥grade 2) and six cases with BN (grade 1) were evaluated as BN with fMRI and follow-up. The radiotracer’s reported diagnostic sensitivity and specificity are both 70–80%, while the fMRI sensitivity and specificity for differentiating from radiation necrosis are 87% and 86%, respectively [26,27]. Though the diagnostic accuracy is superb, the gold standard of BN is still pathological findings. Due to the retrospective nature of the current study, the relatively limited sample, and the difficulty in defining BN, the incidence may be underestimated compared to the gold standard. 

Recently, new radiotracers and techniques have been examined. Radiomics is the field of artificial intelligence that allows the extraction of features from standard bioimaging and the generation of predictive models. Promising results of radiomics have been reported as the definition of potentially radio-resistant tumoral regions and the early identification of recurrence [46]. New radiotracers for discriminating recurrence from BN, F-18-DOPA, or F-18-FET, have been reported [27]. In addition, Cu-64-ATSM is an ATSM-labeled copper isotope used for hypoxia imaging as a monitor of therapeutic response [47]. Cu-64 emits Auger electrons with high linear energy transfer in tissue, which has been demonstrated to induce tumor cell death with high efficiency due to the release of the electrons close to the DNA as well. Thus, this agent is now being studied in a clinical trial (jRCT2091220362) to evaluate its safety and explore the optimal administration dose. 

Other limitations of this study are as follows. Since the study cohort was mainly treated for large BMs, the incidence of BN may have been higher than with small lesions. Furthermore, SRT was delivered with tomotherapy co-planer irradiation. Thus, more conformity and a steeper dose gradient can be provided with other radiation machines, such as gamma knife or cyberknife. In addition, potential biases cannot be excluded from the case–control design. Therefore, a larger prospective registry cohort is needed to address these limitations.

## 5. Conclusions

This comprehensive analysis suggests that larger V55EQD2 cc or V60EQD2 are BN risk factors in five- or ten-fraction SRT. These biologically high doses may affect BN incidence. Thus, the PTV margin should be cut off as much as possible.

## Figures and Tables

**Figure 1 cancers-16-03327-f001:**
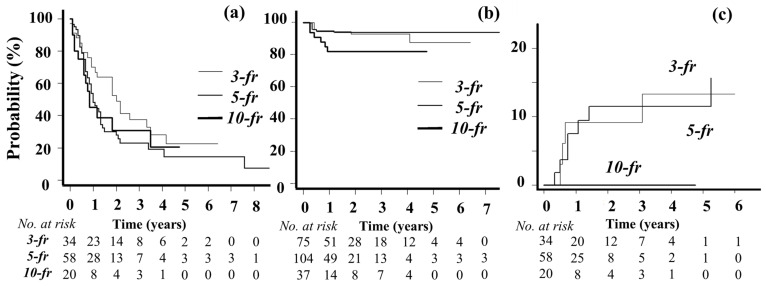
(**a**) Overall survival Kaplan–Meier curves in three-, five-, and ten-fraction groups, (**b**) local control curves of BMs treated with three-, five-, and ten-fraction SRT, and (**c**) the incidence of grade 2 or higher brain necrosis. fr—fraction groups.

**Figure 2 cancers-16-03327-f002:**
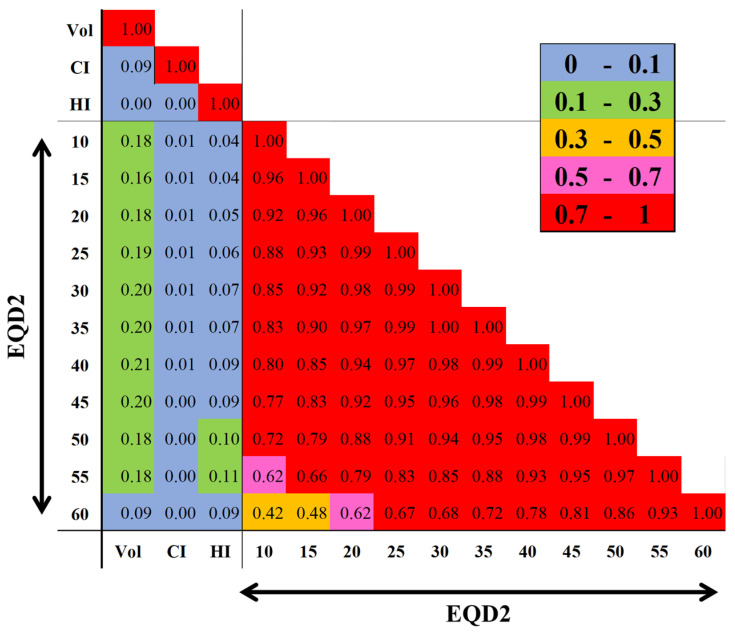
Correlation heatmap of each dosimetric parameter. R^2^s between CI, Vol, UI, and each VxxEQD2 are shown. Colors depicted in the vertical bar on the right side reflect the coefficient of determination. CI, conformity index; R^2^, coefficient of determination; UI, uniformity index; Vol, planning target volume (cc); VxxEQD2, the normal brain volume (cc) receiving an xx Gy biological equivalent dose in 2 Gy fractions.

**Figure 3 cancers-16-03327-f003:**
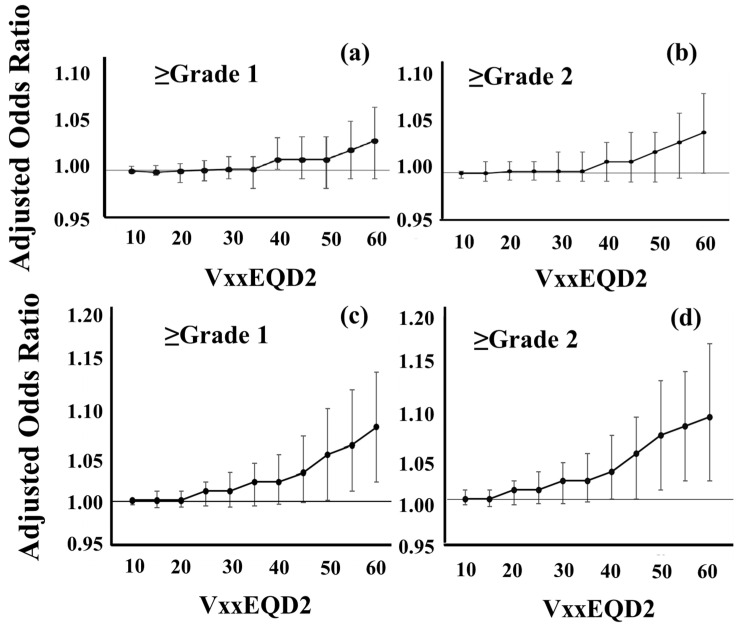
Adjusted odds ratios in logistic regression analyses. In model development, variables included VxxEQD2 (cc), CI, UI, and PTV (cc). Dots indicate the adjusted odds ratio of VxxEQD2 and bars are 95% confidence intervals. (**a**,**b**) The risk of grade 1 or 2 brain necrosis in all 112 patients and (**c**,**d**) the risk after excluding the three-fraction group.

**Figure 4 cancers-16-03327-f004:**
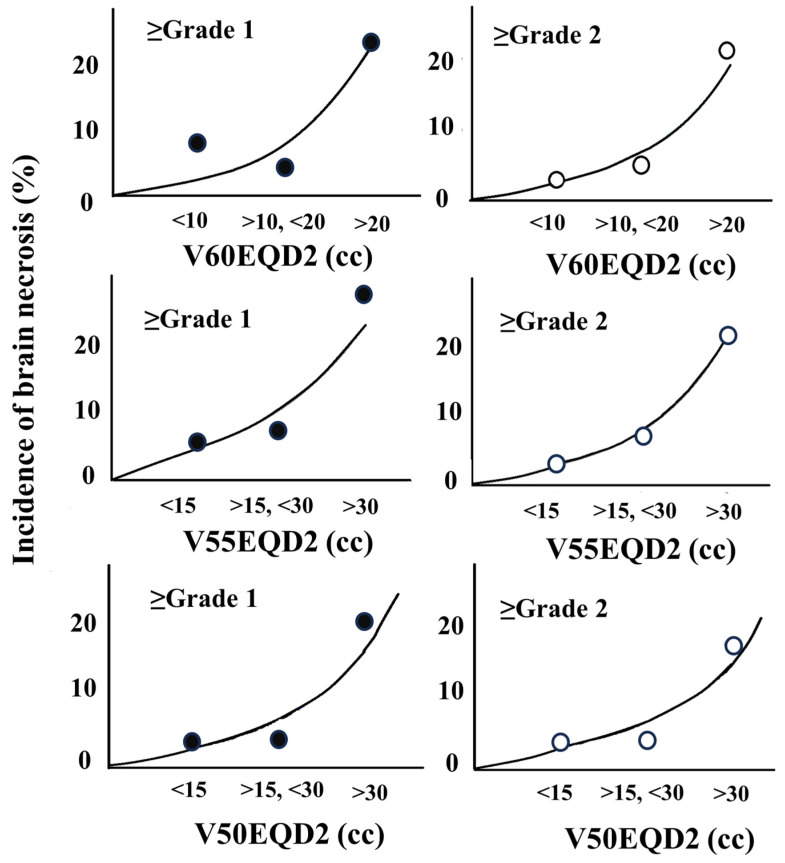
Incidence of brain necrosis (%) in V50EQD2, V55EQD2, and V60EQD2. The incidences of brain necrosis (≥grade 2) were 3% (V60EQD2 ≤10 cc), 5% (>10 cc, ≤20 cc), and 21% (>20 cc) (V60EQD2). Brain necrosis (≥grade 2) occurred in 3% (V55EQD2 ≤10 cc), 8% (>15 cc, ≤30 cc), and 22% (>30 cc) and also in 4% (V50EQD2 ≤15 cc), 4% (>15 cc, ≤30 cc), and 18% (>30 cc) (Appendix A).

**Table 1 cancers-16-03327-t001:** Conversion table in each fraction schedule.

		Biological Equivalent Dose in 2 Gy/fr										
	*fr No*.	10	15	20	25	30	35	40	45	50	55	60
*Dose (Gy)*	*3*	8	10	12	14	16	17	19	20	21	23	24
	*5*	10	14	15	17	20	21	23	25	27	28	30
	*10*	12	16	20	23	26	28	31	33	35	37	40

fr; fraction, No.; number.

**Table 2 cancers-16-03327-t002:** Patient characteristics and treatment details.

Treatment Protocol		30 Gy/3 fr	35 Gy/5 fr	37.5 Gy/5 fr	40 Gy/10 fr
Patient number		34	30	28	20
Age	(mean ± SD)	67.1 ± 10.0	67.4 ± 9.6	63.6 ± 9.3	65.2 ± 16.2
	<60, ≥60 <70, ≥70	13, 7, 15	12, 9, 13	17, 8, 9	16, 6, 5
Sex (female, male)		8, 26	14, 16	13, 15	8, 12
Extracranial disease (+, −)		31, 3	28, 2	28, 0	19, 1
Performance status (0, 1, 2)		6, 25, 3	6, 21, 3	3, 21, 4	6, 7, 7
Primary cancer (patient No.)					
	Lung cancer	29	20	17	10
	GI cancer	2	2	3	3
	Breast cancer	1	1	3	2
	Renal cancer	0	2	2	0
	Sarcoma	0	0	2	1
	Urothelial cancer	1	2	0	1
	Others	1	3	1	3
Total BM No.		75	57	47	37
Median (range)/patient		1.5 (1–9)	1 (1–8)	1 (1–5)	1 (1–6)
	single, multiple	17, 17	19, 11	19, 9	13, 7
Total PTV (cc) (mean ± SD, median)		4.3 ± 4.7, 3.2	7.1 ± 3.7, 6.4	24.0 ± 17.5, 20.2	25.9 ± 13.0, 24.3
Prescribed dose (Gy) *		30 (18–30)	35 (30–35)	37.5 (30–37.5)	40 (36–40)
D95%(Gy) (mean ± SD)		29.2 ± 1.9	33.3 ± 2.1	35.3 ± 3.0	37.7 ± 1.7
D98%(Gy) (mean ± SD)		28.7 ±1.9	32.7 ± 2.2	34.6 ± 3.0	37.0 ± 1.9
D2%(Gy) (mean ± SD)		32.1 ± 2.1	37.0 ± 2.6	38.6 ± 3.5	41.5 ± 2.2
CI (mean ± SD)		3.05 ± 7.13	1.40 ± 1.27	2.34 ± 4.37	1.16 ± 0.63
UI (mean ± SD)		1.09 ± 0.05	1.11 ± 0.09	1.09 ± 0.08	1.09 ± 0.06

Total PTV (cc) showed a significant difference in the four groups (*p* < 0.00001). * (median) (range); BM, brain metastases; CI, conformity index; Dx%, minimum dose delivered to ×% of PTV; GI, gastrointestinal; No., number; UI, uniformity index; PTV, planning target volume; SD, standard deviation.

**Table 3 cancers-16-03327-t003:** Logistic regression analyses of ≥grade 1 or ≥grade 2 brain necrosis in the five- and ten-fraction groups.

**≥Grade 1 Brain Necrosis**			
	**AOR**	**(95% Confidence Interval)**	***p*-Value**
PTV (cc)	0.98	(0.93–1.04)	0.52
V60EQD2 (cc)	1.07	(1.02–1.12)	0.01
	**AOR**	**(95% Confidence Interval)**	***p*-Value**
PTV (cc)	0.98	(0.93–1.04)	0.48
V55EQD2 (cc)	1.05	(1.01–1.1)	0.02
	**AOR**	**(95% Confidence Interval)**	***p*-Value**
PTV (cc)	0.98	(0.93–1.04)	0.49
V50EQD2 (cc)	1.04	(1–1.08)	0.04
**≥Grade 2 Brain Necrosis**			
	**AOR**	**(95% Confidence Interval)**	***p*-Value**
PTV (cc)	0.99	(0.93–1.05)	0.68
V60EQD2 (cc)	1.09	(1.03–1.15)	0.005
	**AOR**	**(95% Confidence Interval)**	***p*-Value**
PTV(cc).	0.99	(0.93–1.05)	0.63
V55EQD2(cc)	1.07	(1.00–1.12)	0.01
	**AOR**	**(95% Confidence Interval)**	***p*-Value**
PTV(cc).	0.99	(0.93–1.05)	0.63
V50EQD2(cc)	1.06	(1.01–1.11)	0.01

AOR, adjusted odds ratio; CI, conformity index; PTV, planning target volume; UI, uniformity index; VxEQD2, normal brain volume irradiated × Gy equivalent dose in 2 Gy/fraction.

**Table 4 cancers-16-03327-t004:** Logistic regression analyses of ≥grade 2 brain necrosis in five- and ten-fraction groups after categorizing variables.

		**AOR**	**(95% Confidence Interval)**	***p*-Value**
PTV (cc)	(<8)	1.00		
	(≥8, <15)	0.15	(0.01–3.18)	0.22
	(≥15)	0.26	(0.02–3.52)	0.31
V60EQD2 (cc)	(<10)	1.00		
	(≥10, <20)	1.81	(0.10–32.1)	0.69
	(≥20)	18.1	(1.14–290)	0.04
		**AOR**	**(95% Confidence Interval)**	***p*-Value**
PTV (cc)	(<8)	1.00		
	(≥8, <15)	0.17	(0.01–3.33)	0.24
	(≥15)	0.19	(0.01–2.94)	0.23
V55EQD2 (cc)	(<15)	1.00		
	(≥15, <30)	4.45	(0.33–59.8)	0.26
	(≥30)	28.7	(1.19–691)	0.04
		**AOR**	**(95% Confidence Interval)**	***p*-Value**
PTV (cc)	(<8)	1.00		
	(≥8, <15)	0.26	(0.02–4.09)	0.34
	(≥15)	0.39	(0.04–3.61)	0.40
V50EQD2 (cc)	(<15)	1.00		
	(≥15, <30)	1.43	(0.08–25.3)	0.81
	(≥30)	9.28	(0.69–126)	0.09

AOR, adjusted odds ratio; PTV, planning target volume; VxEQD2, normal brain volume irradiated × Gy equivalent dose in 2 Gy/fraction.

## Data Availability

Data that support the results of this study are available from the corresponding author, T.M., upon reasonable request.

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
