# Peer review of "Fractionated Stereotactic Intensity-Modulated Radiotherapy for Large Brain Metastases: Comprehensive Analyses of Dose–Volume Predictors of Radiation-Induced Brain Necrosis"

_cancers, 2024, doi:10.3390/cancers16193327_

Round 1

Reviewer 1 Report

Comments and Suggestions for Authors

Murai et al performed a retrospective analysis of patients treated with stereotactic radiotherapy for brain metastasis at a single institution between 2012 and 2021 were treated with up to 10 metastases.. Based on the retrospective analysis, the authors determined that the incidence of symptomatic necrosis was higher in patients with a larger biological doses of 50, 55 or 60 Gy and was significantly higher in patients with greater than 30 mL volume at 55 gray and 20 mL for 60 gray, with a conclusion that volumes should be kept lower than this to lower the risk of brain necrosis.

Simple summary: Provided a brief summary of the protocol followed in the retrospective review and results.

Abstract: The abstract is not much longer than the simple summary and provided the same information. However, no changes recommended for either.

Keywords: Appropriate to the submission.

Introduction: The introduction provided background information including the lack of information regarding predictors of necrosis that had not been established in 5 and 10 faction stereotactic radiotherapy.

Materials and methods: In 2.1 study design and patients, exclusion criterion number 5 should use the term iodine or gadolinium contrast. In the next sentence, it was noted "informed consent was obtained from all patients or their guardians." It is unclear whether this was informed consent for the radiation therapy that was performed or whether this was specific consent for the retrospective study.  2.2 Radiotherapy details & 2.3 patient follow-ups and dose-volume parameters describe the treatments provided to the patients. This is well-written, understandable, and well complemented by table 1. In line 100, “fMRI was indicated to represent “perfusion and functional MRI.”  The term functionnal MRI needs to be defined.  In 2.4 statistical analysis, no obvious concerns found.

Results: Section 3.1. Patient characteristics, treatment details, and outcomes. Patient characteristics were well summarized in table 2 that supplements the patient characteristics like section 3.1.

3.2 toxicities. There was a grade 5 toxicity documented in lines 158-159. The authors should confirm that the cause of death grade 5 toxicity was the bleeding and not something else. In table 3, under "Grade 2 brain necrosis", does this refer only to grade 2 or grade 2 and above brain necrosis?  Lines 199-204 are best discussed as a summary of an organized table and does not read well as a narrative.

Discussion: the discussion adequately addresses the results of this retrospective analysis and their implications. Limitations of the study were adequately discussed.

Conclusions: the conclusions should not make definitive recommendations for radiation dosing based on volume given the retrospective analysis of this study. These need to be more forcefully documented as recommendations pending additional prospective studies.

References: all references are adequate and pertain to the study

Supplementary materials: adequate to provide supplemental information to better understand the narrative in the results section.

Author Response

We appreciate the favorable comments of reviewers. Our responses are shown below. The edited parts are highlighted in red.

[Responses to Reviewer #1]

Revision 1:

The reviewer stated “Materials and methods: In 2.1 study design and patients, exclusion criterion number 5 should use the term iodine or gadolinium contrast.”

Thank you for your suggestion. We have added the phrase "contrast." (2. Materials and Methods, 2.1 Study design and patients, Line 8th).

Revision 2:

The reviewer suggested we clarify “informed consent” in Materials and methods, 2.1 Study design and patients.

Thank you for your suggestion. This study design is the retrospective observation of a population treated with prospective protocols. The consent is acquired for the treatment, not for this study. We have added the phrase “for the SRT” (2. Materials and Methods, 2.1 Study design and patients, Line 8th -9th ).

Revision 3:

The reviewer stated, “The term functional MRI needs to be defined.”

Yes. We should have explained the functional MRI details. Functional MRI means MR spectroscopy and MR perfusion images. The functional MRI was perfusion and magnetic resonance spectroscopy. We have rephrased the paragraphs as “BN was pathologically verified or diagnosed by perfusion and magnetic resonance spectroscopy (fMRI) or C-11 methionine positron emission tomography (MetPET) [19-21]. Recurrence in MetPET was diagnosed based on the mean tumor/cortex ratio values (> 2.0). Lesions with N-acetyl aspartate/choline ratio < 1 or lipid/choline ratio < 3 in the fMRI or relative cerebral blood volume of > 2.1 were considered recurrence. These were diagnosed by diagnostic radiologists with >5 years of neuroradiology experience. Other than recurrence was defined as BN.” (2. Materials and Methods, 2.3 Patient follow-ups, endpoints, and dose-volume parameters, Para 2).

Revision 4:

The reviewer suggested we should confirm the cause of death of the grade 5 intracranial hemorrhage.

Yes. We confirmed the cause of death of the patient. This patient was brought to the emergency department in our hospital with a loss of consciousness. The brain CT showed intratumor bleeding with brain herniation irradiated with SRT. Decompressive craniectomy was not indicated considering the patient’s cancer status. Thus, we evaluated the bleeding caused by SRT.

We have added the sentences, “The brain CT of the grade 5 patient showed intratumor bleeding with brain herniation irradiated with SRT. Thus, the bleeding was evaluated as caused by SRT.” (3. Results, 3.2. Toxicities, Para 1, Line 2nd – 4th)

Revision 5:

The reviewer asked “In table 3, under "Grade 2 brain necrosis", does this refer only to grade 2 or grade 2 and above brain necrosis?”

Yes. It is grade 2 or more brain necrosis. We have rephrased the phrase grade in Tables 3 and 4.

Revision 6:

The reviewer stated “paragraph 3 in 3.3. Multivariate analyses of BN in 3. Results are best discussed as a summary of an organized table”.

Thank you for your suggestion. We have revised the paragraph and added Table S2 summarizing Figure 3.

Revision 7:

The reviewer suggested “The conclusions should not make definitive recommendations for radiation dosing based on volume given the retrospective analysis of this study. These must be more forcefully documented as recommendations pending additional prospective studies.”

Thank you for your suggestion. We have revised the conclusions: "This comprehensive analysis suggests that larger V55EQD2 cc or V60EQD2 are BN risk factors in 5- or 10-fraction SRT. These high biologically high doses may affect BN incidence. Thus, the PTV margin should be cut off as much as possible.”

We have also rephrased the conclusion in Abstract.

Reviewer 2 Report

Comments and Suggestions for Authors

I would thank the Editors for the opportunity to review this manuscript. The topic is interesting and the paper offers new data that could aid to identify dosimetric predictors of radiation necrosis. Nonetheless, in order to be suitable for publication I would suggest the following revisions.

Introduction

"radiotherapies" should be replaced with "radiotherapy" in the first sentence and "radiotherapy techniques" in the second

Sentences should not start with abbreviations (e.g. BN)

VxxGyE could be replaced with "EQD2"?

Dosimetric predictors of brain necrosis (BN) have not yet been established” Actually, in addition to the papers by Inoue et al. And Andreaska et al. cited in the discussion, other studies proposed dosimetric predictors of RN for SRT in 3 fractions (https://doi.org/10.1016/j.ijrobp.2023.07.040 https://doi.org/10.1016%2Fj.ijrobp.2020.08.013) and in 5 fractions, as described also in Hytec report https://doi.org/10.1016%2Fj.ijrobp.2020.10.039

Material and methods

"PTV denoted a visible tumor on computed tomography (CT)..." This sentence should be reformulated for clarity. I suggest to describe image fusion, how GTV was defined and then expansion to PTV

"the basic prescribed dose was 35 Gy/5 fractions (fr). A dose of 30 Gy/3 fr was permitted for small lesions (maximum diameter <1.5 cm), while 37.5 Gy/5 fr was used for large lesions (≥3 cm)" on this basis is not really clear to me the subset of patients treated with 35 Gy vs those treated with 37.5 Gy.

"intensity-modulated irradiation technique was applied. SRT was delivered with tomotherapy." Please better specify the technique adopted (helical tomotherapy?) and detail it briefly.

"Any systematic agent was not allowed around the time of these treatments" please better specify allowed interval between systemic treatment and RT

"Patients were regularly followed up" could you please define timing of the follow up (both clinical and radiologic)?

"Local recurrence was defined as a ≥20% increase in the maximum diameter" is it defined by an internal protocol? As per RECIST criteria the sum of diameters should be considered. Similarly, I suggest to specify criteria used to define radionecrosis

Results

I suggest to specify also median lesions' volume, as considered mean values and the fact that only 32 lesions were ≥15 cc, there were likely a few very bulky lesions that impacted mean values

"cerebrum" should be likely replaced with "cerebellum"

Could you please explain the difference in CI between the 3 fr group and the other two groups?

Please further explain the decision to exclude the 3 fr group.

It should be specified that, although statistically significant, odds ratio were relatively low for the volume predictors identified

Discussion

"With improvements in the prognosis of patients with advanced-stage cancer, the incidence of BN is recognized as the dose-limiting toxicity of SRT for BMs"

I would split the two concepts (survival improvement and expanded adoption of SRT and SRT also allowed by new techniques, with the emergence of RN as a DLT). The two paragraphs should be expanded, adding references.

The discussion section could as well include clinical aspects of BN ( https://doi.org/10.3390%2Fcancers15092432 ) new perspectives regarding assessment of BN, such as modern imaging techniques and adoption of radiomics to discriminate BN from disease progression and identify potential predictors ( DOI: 10.1016/j.tranon.2021.101275 DOI: 10.3390/cancers15072184 )

"SRT was delivered with tomotherapy co-planer irradiation" please expand the paragraph and better explain this limitation

Among limitations, add the retrospective nature of the study, the relatively limited sample and the difficulty to define RN with relatively low rates

Comments on the Quality of English Language

English requires minor revisions

Author Response

We appreciate the favorable comments of reviewers. Our responses are shown below. The edited parts are highlighted in red.

[Responses to Reviewer #2]

Revision 1:

The reviewer suggested that "radiotherapies" should be replaced with "radiotherapy" in the first sentence and "radiotherapy techniques" in the second” in Introduction.

Thank you for the suggestions. We have rephrased these phrases in Introduction (Para 1, lines 2nd and 3rd).

Revision 2:

The reviewer stated that sentences should not start with abbreviations (e.g. BN).

We have rephrased the phrase in Lines 5th and 7th, Para 2, Introduction.

Revision 3:

The reviewer stated “VxxGyE could be replaced with "EQD2"?”

Yes. We have rephrased “VxxGyE” to “VxxEQD2” in this manuscript.

Revision 4:

The reviewer stated that other studies proposed dosimetric predictors of brain necrosis for SRT in 3 fractions and 5 fractions, as described also in the HyTEC report.

              We agree with the recommendations. We have revised the introduction and discussion. We have added references as the reviewers recommended and rephrased sentences in Introduction and Discussions, stated the clinical importance in this immunotherapy and targeted therapy era, and emphasized the novel aspects of this study.

Revision 5:

The reviewer suggested describing image fusion, how GTV was defined and then expansion to PTV in Material and methods.

We have revised the 2nd paragraph in 2.2. Radiotherapy details, in Material and methods, as “Computed tomography (CT) was taken and fused with contrast-enhanced magnetic resonance imaging (MRI). The sequence was axial T1-weighted imaging with 1 mm slice-thickness mainly taken with 1.5-T machines. The clinical target volume was defined as an abnormal contrast-enhanced lesion on the MRI. The planning target volume (PTV) was the target plus 0-2 mm margin depending on the nearby critical structures, such as the optic nerve, brain stem, and so on.”

Revision 6:

The reviewer suggested describing the subset of patients treated with 35 Gy vs those treated with 37.5 Gy.

Thank you for the suggestion. We have separated the 5-fraction group into 2 in Table 1. Statistical differences were examined in these 4 groups again.

Revision 7:

The reviewer suggested specifying the technique adopted and detailing it briefly.

Yes. We applied helical tomotherapy to SRT and the stimulating integrated boost technique is used to reduce critical organ doses in 2.2. Radiotherapy details, in 2. Materials and Methods.

We have added sentences in 3rd paragraph in 2.2. Radiotherapy details, in 2. Materials and Methods, as “. To satisfy these limitations, the stimulating integrated boost technique is used to reduce critical organ doses. SRT was delivered with helical tomotherapy.”

Revision 8:

The reviewer suggested specifying the allowed interval between systemic treatment and SRT.

Thank you for the suggestion. The acceptable interval between SRT and systematic therapy is generally 1 week before and after SRT. We have added a sentence (2. Materials and Methods,.2.2. Radiotherapy details, Para 3, Line 5-7.)

Revision 9:

The reviewer suggested defining the timing of the follow-up (both clinical and radiologic).

Yes. All patients were followed every 2 to 3 months after SRT with contrast-enhanced MRI and physical examination. We have rephrased the 1st sentence in 2.3 Patient follow-ups, endpoints, and dose-volume parameters, in 2. Materials and Methods.

.

Revision 10:

The reviewer stated "Local recurrence was defined as a ≥20% increase in the maximum diameter", Is it defined by an internal protocol? As per RECIST criteria, the sum of diameters should be considered. Similarly, I suggest specifying criteria used to define radio-necrosis.”

Sorry. The previous description “maximum diameter” was incorrect based on RECIST 1.1. We have rephrased the sentence in 1st paragraph in 2.3 Patient follow-ups, endpoints, and dose-volume parameters, in 2. Materials and Methods, as “Local recurrence was defined as a ≥20% increase in the sum of diameters of target lesions of the contrast-enhanced tumor on MRI or CT, taking as reference the smallest sum on study.”

 We have revised the 2nd paragraph in 2.3 Patient follow-ups, endpoints, and dose-volume parameters, in 2. Materials and Methods, as “Radiation necrosis was diagnosed pathologically or MRI. The MRI was evaluated by independent imaging review by diagnostic radiologists with >5 years of neuroradiology experience. We have added sentences such as “Recurrence in MetPET was diagnosed based on the mean tumor/cortex ratio values (> 2.0). Lesions with N-acetyl aspartate/choline ratio < 1 or lipid/choline ratio < 3 in the fMRI or relative cerebral blood volume of > 2.1 were recalled as recurrence. These were diagnosed by diagnostic radiologists with >5 years of neuroradiology experience. Other than recurrence was defined as brain necrosis.”

Revision 11:

The reviewer also suggested specifying median lesions' volume, as considered mean values and the fact that only 32 lesions were ≥15 cc, there were likely a few very bulky lesions that impacted mean values.

Yes. In this study, the total PTV(cc) of each patient was examined in the multivariate analyses. To clarify the variance, we rephrase the sentence in the 6th line, in 3.1. Patient characteristics, treatment details, and outcomes, in 3. Results. We have added the median lesions’ volume in Table 2. In addition, we have revised the 2nd paragraph in 2.2. Radiotherapy details, in 2. Materials and Methods, “Since SRT for large total PTV (≥15 cc) was associated with a risk of neurotoxicity, we employed different fractionation protocols for these tumors [19]. Between 2017 and 2021, the protocol for large total PTV was revised to reduce the incidence of BN. Treatment for large lesions (total PTV(cc) ≥15 cc) with 35 Gy or 37.5 Gy/5 fr was superseded by 40 Gy/10 fr. In the new protocol, 30 Gy/3 fr for small lesions and 35 Gy/ 5 fr for other lesions remained.”

Revision 11:

The reviewer suggested "cerebrum" should be likely replaced with "cerebellum"

Yes. We have rephrased "cerebrum" to "cerebellum" in the 9th line, in the 1st paragraph in 3.1. Patient characteristics, treatment details, and outcomes, in 3. Results.

Revision 12:

The reviewer suggested explaining the difference in CI between the 3 fr group and the other two groups.

The CI in the 3-fr group varied widely. It may be that 3-fraction was applied to smaller lesions. We have added a sentence “As the 3-fraction was used for smaller lesions, the CI in the 3-fraction’s variance was wide but not significant (p > 0.36).” in the last line, in the 1st paragraph in 3.1. Patient characteristics, treatment details, and outcomes, in 3. Results.

Revision 13:

The reviewer suggested further explaining the decision to exclude the 3 fr group.

Thank you for your suggestion. The linear-quadratic model fits well if a single-fraction dose was less than 2-fold of the organ α/β ratio. In this study, the α/β ratio of the normal brain was assumed as 2, and the irradiated doses per fraction to the organ were almost 6-7 Gy in 3-fr, 4-5 Gy in 5-fr, and 2-3 Gy in 10-fr, respectively. Therefore, the calculated biological equivalent dose in the 3-fraction group may not correspond biologically to that in the 5- or 10-fraction group. Besides, in this study, the population in the 3-fr group was 34, and not sufficient for independent multivariate analyses to explore the risk factors of brain necrosis. Thus, biological equivalent dose calculation is necessary to address this issue.

We have revised the 3rd paragraph in 2.4. Statistical analysis in in 2. Materials and Methods.

In addition, we have added sentences in

Revision 14:

The reviewer suggested specifying that, although statistically significant, odds ratios were relatively low for the volume predictors identified.

Yes. We have added sentences in the 2nd paragraph of 3.3. Multivariate analyses of BN, in 3. Results, as “The 3-fraction group was excluded from these LRA due to the biological imprecision of EQD2 in the fractionation. Besides, the sample size of the group was not sufficient for independent analyses. Therefore, the final LRA was conducted after excluding the 3-fraction group.”

Revision 15:

The reviewer suggested splitting the two concepts: 1) survival improvement and expanded adoption of SRT and SRT also allowed by new techniques, with the emergence of RN as a DLT) and expand the two paragraphs and add references in the 1st paragraph, in 4. Discussion.

We agree with this suggestion. We have revised the discussion. In the 1st paragraph, we have stated current advances in systematic therapy including immunotherapy; e.g. immune checkpoint inhibitors and targeted therapy. Interaction between these new immunotherapy agents and SRT has not been established. Thus, we have added a sentence “dose constraints in SRT should be more prudent and conservative in this immuno-therapy era.” 

 We have added the 2nd paragraph to explain the expansion of the SRT role including peri-operative irradiation, and multiple or larger targets.

Revision 16:

The reviewer stated “The discussion section could also include clinical aspects of brain necrosis and new perspectives regarding assessment of BN, such as modern imaging techniques and adoption of radiomics to discriminate BN from disease progression and identify potential predictors.

We agree with this suggestion. We have added the 7th paragraph in 4. Discussion to explain the discrimination between recurrence and brain necrosis. In the next 8th paragraph, we explain the details of diagnostic radiotracers and theranostics agent of ATSM.

Revision 17:

The reviewer stated "SRT was delivered with tomotherapy co-planer irradiation. Please expand the paragraph and better explain this limitation.”

Thank you for the suggestion. We have added sentences in the last paragraph in 4. discussion, as “more conformity and a steeper dose gradient can be provided with other radiation machines, such as gamma knife or cyberknife.”

Revision 18:

The reviewer stated “Among limitations, add the retrospective nature of the study, the relatively limited sample and the difficulty to define brain necrosis with relatively low rates”

Thank you for the suggestion. We have added a sentence in the 7th paragraph in 4. Discussion, as “Due to the retrospective nature of the current study, the relatively limited sample, and the difficulty in defining BN, the incidence may be underestimated compared to the gold standard.”

Reviewer 3 Report

Comments and Suggestions for Authors

The Article submitted by Murai et al “ Fractioned Stereotactic Intensity-Modulated Radiotherapy for Large Brain Metastases: Comprehensive Analyses of Dose-Volume Predictors of Radiation- Induced Brain Necrosis” is well documented study. My suggestions for Authors,

1) Introduction is shorter. Authors should revise intro part & add more recent information.

2) Ethics protocol approval detail & number should be provided in the MS.

3) In table 2, add demographic deatils also.

4) Coorelation analysis heatmap should provided with r2 value & little description.

5) In table 3 authors should clarofy that Odds ratio is adjusted odds ratio? If not authors should do the Boeferroni’s correction.

Author Response

We appreciate the favorable comments of reviewers. Our responses are shown below. The edited parts are highlighted in red.

[Responses to Reviewer #3]

Revision 1:

The reviewer stated “Introduction is shorter. Authors should revise the intro part & add more recent information.”

Yes. We agree with the comment. We have revised the introduction section and added new references. The revised introduction stated the clinical importance of this immunotherapy and targeted therapy era and emphasized the novel aspects of this study, such as multicollinearity, comprehensive analyses and fractionated SRT for large lesions.

Revision 2:

The reviewer suggested providing ethics protocol approval details & numbers in the MS.

Yes. We agree with the comment. We have added the information about the Institutional Review Board and number (No. 60-19-0207) in 2.1 Study design and patients, in 2. Materials and Methods.

Revision 3:

The reviewer stated, “In table 2, add demographic details also.”

Thank you for the suggestion. We have separated the 5-fraction group into 2 in Table 1. Statistical differences were examined in these 4 groups again.

In addition, we have added patients’ demography (- 60 vs 60-70 vs 70- years old)

Revision 4:

The reviewer stated “4) Correlation analysis heatmap should be provided with r2 value & little description.

We agree with the comment. We have recalculated the coefficient of determination (R2) instead of Pearson’s coefficient and rephrased “absolute Pearson’s coefficient (PCC)” to “coefficient of determination (R2)” and have corrected the heatmap in Figure 2

Revision 5:

The reviewer stated, “In table 3 authors should clarify that odds ratio is adjusted odds ratio?”

Yes. It is an adjusted odds ratio. We have rephrased “odds ratio” to “adjusted odds ratio (AOR)”

Reviewer 4 Report

Comments and Suggestions for Authors

The authors presented a paper about “Fractionated Stereotactic Intensity-Modulated Radiotherapy for Large Brain Metastases: Comprehensive Analyses of Dose-Volume Predictors of Radiation-Induced Brain Necrosis”.

The topic is of particular interest because it deals with a clinical situation which is frequently encountered.

It would interesting to have some additional information such as:

1)      Which kind of magnetic resonance imaging was used to contour the brain metastases?

2)      Was the choice to opt for different fractionation motivated by the overall volume of treatment of all the metastases?

3)      Which is the additional margin used for PTV?

4)      It would be important to have additional details about the prescription isodose (in particular if there were any differences among the different fractionation schedules)?

5)      Which kind of linac was used to treat the patients?

6)      Which kind of reporting scale was used to identify adverse events (CTCAE, RTOG etc)?

7)      Was SRT delivered in combination with chemotherapy or with immunotherapy? If so which kind of immunotherapy was used?

Author Response

We appreciate the favorable comments of reviewers. Our responses are shown below. The edited parts are highlighted in red.

[Responses to Reviewer #4]

Thank you for your favorable comments and recommendations.

Revision 1:

The reviewer stated, “Which kind of magnetic resonance imaging was used to contour the brain metastases?”

Thank you for your comment. We have added sentences in the 2nd paragraph in 2.1 Study design and patients, in 2. Materials and Methods, as “Computed tomography (CT) was taken and fused with contrast-enhanced magnetic resonance imaging (MRI). The sequence was axial T1-weighted imaging with 1 mm slice-thickness mainly taken with 1.5-T machines. The clinical target volume was defined as an abnormal contrast-enhanced lesion on the MRI. The PTV was the target plus 0-2 mm margin depending on the nearby critical structures, such as the optic nerve, brain stem, and so on.”

Revision 2:

The reviewer stated “Was the choice to opt for different fractionation motivated by the overall volume of treatment of all the metastases?

Yes. The fractionation schedule was decided depending on the overall volume (total PTVcc). To clarify these, we have revised the 2nd paragraph in 2.1 Study design and patients, 2. Materials and Methods.

Revision 3:

The reviewer stated, “Which is the additional margin used for PTV?”

We should have explained the treatment in detail. The shorter margin was applied based on critical organs nearby targets. To reduce the doses to the tolerance level, the margin was cut.

We have added sentences in the 2nd paragraph of 2.2. Radiotherapy details, 2. Materials and Methods, as “The PTV was the target plus 0-2 mm margin depending on the nearby critical structures, such as the optic nerve, brain stem, and so on. These margins were cut in cases where critical organs were nearby targets.”

Revision 4:

The reviewer recommended adding details about the prescription of isodose.

All doses were prescribed to D50 where the prescribed dose covers 50% of the target volume.

We have added sentences in the 2nd paragraph of 2.2. Radiotherapy details, 2. Materials and Methods, as “The prescribed dose covered at least 50% of PTV. At least 90% of the dose covered 95% of the PTV.”

Revision 5:

The reviewer recommended adding details about the treatment machine.

Thank you for the comment. We used Hi-Art system or Radixact. Helical tomotherapy mode was applied to all treatments.

We have added a sentence in the 3rd paragraph of 2.2. Radiotherapy details, 2. Materials and Methods, as “The SRT was delivered with helical mode in tomotherapy, Hi-ART system® or Radixact®.”

Revision 6:

The reviewer stated, “Which kind of reporting scale was used to identify adverse events (CTCAE, RTOG, etc)?”

We have used CTCAE v4 criteria. We have added the phrase “CTCAE” in the 3rd paragraph in 2.3 Patient follow-ups, endpoints, and dose-volume parameters, 2. Materials and Methods,

Revision 7:

The reviewer stated “Was SRT delivered in combination with chemotherapy or with immunotherapy? If so which kind of immunotherapy was used?”

Yes. We did not allow the combinations. We have added a sentence in the 3rd paragraph of 2.2. Radiotherapy details, 2. Materials and Methods, as “Any systematic agent, including immunotherapy and tyrosine kinase inhibitors (TKI), was not allowed 1 week before and after these treatments to avoid increased toxicity.

Round 2

Reviewer 2 Report

Comments and Suggestions for Authors

Most of the suggested revisions have been addressed, improving the quality of tha paper.

Nonetheless, I would suggest a further proofreading as English still needs to be improved. Moreover, I suggest the following corrections

Simple summary and abstract

cut off as much as possible” is not really clear and should be rephrased

Introduction

line 57 “concurrent systemic therapies[4]. Including” remove the point and replace it with a comma

While, few studies have …” remove while

Dose-volume predictors in fractionated SRT have been reported by fewer than radiosurgery” rephrase as “fewer studies compared with radiosurgery”

Immunotherapy or targeted therapy improves overall survival significantly [16-18].” expand this concept

Methods

stimulating integrated boost” should be “simultaneous integrated boost”?

taken” replace with “acquired”

Other than recurrence was defined as BN” please rephrase and clarify

Comments on the Quality of English Language

I would suggest a further proofreading as English still needs to be improved. 

Author Response

We appreciate the favorable comments of reviewers. Our responses are shown below. The edited parts are highlighted in red.

Besides, we have rephrased captions “Odds Ratio” in Figure 3 to “Adjusted Odds Ratio”.

Revision 1

The reviewer stated that “cut off as much as possible” is not clear and should be rephrased in Simple summary and abstract.

Thank you for your suggestion. We have rephrased sentences as “minimize the margin as much as possible”. We have replaced other “cut-offs” as “minimize” as well.

Revision 2

The reviewer stated “line 57 “concurrent systemic therapies[4], including” remove the point and replace it with a comma” in Introduction.

We have rephrased sentences as “concurrent systemic therapies, including immunotherapy or targeted therapy.”.

Revision 3

The reviewer stated that  “While, few studies have …” remove "while".

We have removed “While”

Revision 4

The reviewer stated “Dose-volume predictors in fractionated SRT have been reported by fewer than radiosurgery” rephrased as “fewer studies compared with radiosurgery”

We have rephrased the sentence: "Dose-volume predictors in fractionated SRT have been reported by fewer studies compared with radiosurgery”.

Revision 5

The reviewer stated “Immunotherapy or targeted therapy improves overall survival significantly [16-18].” expand this concept”

We agree with the recommendations. We have revised the 3rd paragraph in the introduction.

We have added sentences on molecular agents and immunotherapy and described the 2 examples of clinical trials, DESTINY-Breast04 and CheckMate 067 trials.

Revision 6

The reviewer stated “stimulating integrated boost” should be “simultaneous integrated boost” in Methods.

Yes. It is mistyping. We have rephrased “stimulating integrated boost” as “simultaneous integrated boost.”

Revision 7

The reviewer stated that “taken” was replaced with “acquired” in Methods.

We have rephrased “taken” as “acquired”.

Revision 8

The reviewer stated that “Other than recurrence was defined as BN” should be rephrased and clarified”

We have rephrased the sentence as “If no radiological or pathological evidence of recurrence is observed, the lesion can be classified as BN.”

Reviewer 4 Report

Comments and Suggestions for Authors

I have no additional comments

Author Response

We appreciate the favorable comments.

We have rephrased the captions “Odds Ratio” in Figure 3 to “Adjusted Odds Ratio”.